# Constructing an adult orofacial premotor atlas in Allen mouse CCF

Jun Takatoh[1,2]*[†], Jae Hong Park[3†], Jinghao Lu[2†], Shun Li[2], PM Thompson[3], Bao-Xia Han[2], Shengli Zhao[2], David Kleinfeld[4,5], Beth Friedman[6], Fan Wang[1,2,3]*

[1]Department of Brain and Cognitive Sciences, Massachusetts Institute of Technology, Cambridge, United States; [2]Department of Neurobiology, Duke University, Durham, United States; [3]Department of Biomedical Engineering, Duke University, Durham, United States; [4]Section of Neurobiology, University of California at San Diego, San Diego, United States; [5]Department of Physics, University of California at San Diego, San Diego, United States; [6]Department of Computer Science and Engineering, University of California at San Diego, San Diego, United States

**Abstract** Premotor circuits in the brainstem project to pools of orofacial motoneurons to execute essential motor action such as licking, chewing, breathing, and in rodent, whisking. Previous transsynaptic tracing studies only mapped orofacial premotor circuits in neonatal mice, but the adult circuits remain unknown as a consequence of technical difficulties. Here, we developed a three-step monosynaptic transsynaptic tracing strategy to identify premotor neurons controlling vibrissa, tongue protrusion, and jaw-closing muscles in the adult mouse. We registered these different groups of premotor neurons onto the Allen mouse brain common coordinate framework (CCF) and consequently generated a combined 3D orofacial premotor atlas, revealing unique spatial organizations of distinct premotor circuits. We further uncovered premotor neurons that simultaneously innervate multiple motor nuclei and, consequently, are likely to coordinate different muscles involved in the same orofacial motor actions. Our method for tracing adult premotor circuits and registering to Allen CCF is generally applicable and should facilitate the investigations of motor controls of diverse behaviors.

*For correspondence:
jtakatoh@mit.edu (JT);
fan_wang@mit.edu (FW)

[†]These authors contributed equally to this work

**Competing interest:** The authors declare that no competing interests exist.

## Introduction

Motor actions are the primitive motor commands that are coordinated to produce behavior (*Tinbergen, 1951*). Orofacial motor actions, which underlie breathing, drinking, and eating, are essential for animals to access vital sustenance, that is, oxygen, water, and food (*Moore et al., 2014*). For example, water can be consumed by a sequence of orofacial motor actions, including jaw opening, licking, and swallowing. Mice, as nocturnal animals, use coordinated whisking and sniffing, often in coordination with breathing, to explore their physical environment (*Deschênes et al., 2012*; *Kleinfeld and Deschenes, 2011*; *Moore et al., 2014*; *Welker, 1964*). Thus, many of the orofacial behaviors utilize multiple muscles through coordinated activity of their associated motoneurons in a seamless manner (*Kurnikova et al., 2017*; *McElvain et al., 2018*; *Moore et al., 2014*). Given that each orofacial motoneuron pool only projects to its corresponding muscle and lacks collaterals to innervate other central neurons, and that individual motor actions may involve multiple muscles, coordinated orofacial behaviors are thought to be achieved by orofacial premotor circuits in the brainstem (*Moore et al., 2014*). Thus, to understand how these brainstem circuits integrate information of external sensory stimuli, self-motion signals, and animals' needs or internal states to orchestrate the activities of multiple, distinct groups of cranial motoneurons, a key first step is to delineate the orofacial premotor circuit for each group of motoneurons in the adult nervous system. Putative premotor neurons that

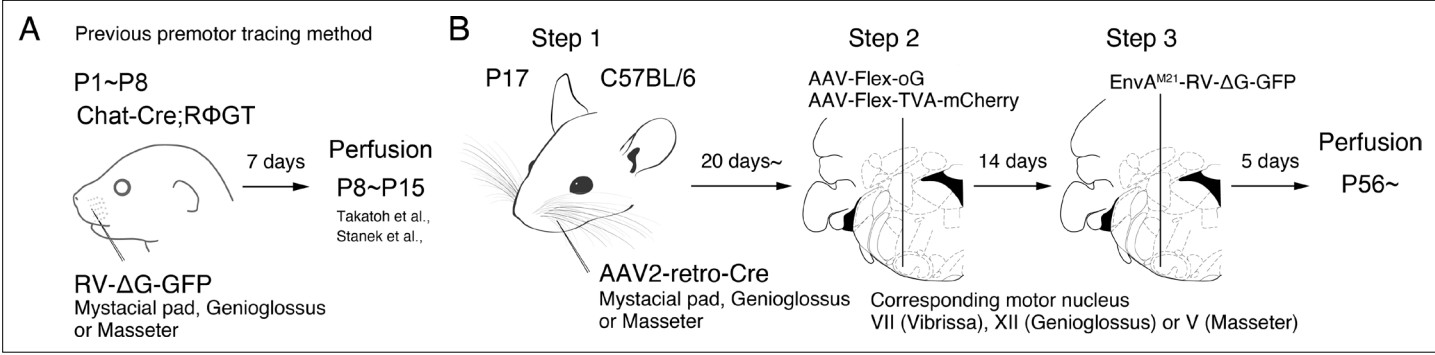

**Figure 1.** Monosynaptic rabies virus tracing strategy for labeling adult orofacial premotor circuits. (**A**) Schematic of previously used monosynaptic premotor transsynaptic tracing method in neonatal mice. (**B**) Schematic of the three-step monosynaptic premotor tracing strategy in adult mice developed in this study.

The online version of this article includes the following figure supplement(s) for figure 1:

**Figure supplement 1.** Optimization of the timing of AAV2-retro-Cre injection in the peripheral muscle.

project to different craniofacial motor nuclei had been previously mapped using conventional retrograde tracers injected into different motor nuclei in the brainstem (*Borke et al., 1983*; *Hattox et al., 2002*; *Isokawa-Akesson and Komisaruk, 1987*; *Travers and Norgren, 1983*; *Vornov and Sutin, 1983*). However, each orofacial motor nucleus contains functionally different and also antagonistic motor neurons (*Aldes, 1995*; *Ashwell, 1982*; *Klein and Rhoades, 1985*; *Krammer et al., 1979*; *Mizuno et al., 1975*). For instance, the hypoglossal nucleus contains motoneurons for tongue protrusion, retrusion, and shaping. Injection of conventional retrograde tracer into the hypoglossal nucleus would, therefore, label premotor circuits for all these groups of muscles (*Aldes, 1995*; *Gestreau et al., 2005*; *Krammer et al., 1979*). Moreover, injection of conventional tracer could label neuronal populations projecting to non-motoneurons (e.g., interneurons) adjacent to the injection site. Thus, a major limitation of conventional tracers is the lack of muscle/motoneuron specificity. An improvement is the injection of transsynaptic viral tracers into specific muscle groups, such as replication-competent rabies (*Kurnikova et al., 2019*) and pseudorabies (*Mercer Lindsay et al., 2019*). This approach identifies presynaptic nuclei but suffers from the uncertainty of labeling higher order in addition to presynaptic nuclei.

To overcome such limitations, we and others used a monosynaptic retrograde rabies virus transsynaptic tracing strategy and examined the premotor circuits for various craniofacial and somatic motoneurons in neonatal mice (*Sreenivasan et al., 2015*; *Stanek et al., 2014*; *Stepien et al., 2010*; *Takatoh et al., 2013*; *Tripodi et al., 2011*). In this strategy, the specific muscle of interest is inoculated with a glycoprotein (G protein)-deleted RV (ΔG-RV) that encodes a fluorescent protein, and ΔG-RV is taken up by motor axon terminals in the muscle and retrogradely transported to the motoneuron cell bodies. ΔG-RV is trans-complemented with G protein in motoneurons and subsequently spread into premotor neurons. The lack of G protein in premotor neurons prevents further retrograde traveling of ΔG-RV (*Figure 1A*). However, ΔG-RV does not infect from peripheral muscles efficiently in animals older than P8, thus precluding the use of this strategy to trace premotor circuits beyond approximately P15 (*Takatoh et al., 2013*). Since many orofacial behaviors do not fully develop until after weaning (*Westneat and Hall, 1992*), it is uncertain whether the orofacial premotor circuits revealed for neonatal mice are the same in adult animals, at age at which most of the behavior and electrophysiology experiments are conducted.

Another previously unsolved issue is to map traced premotor neurons from different muscles to a standard reference frame to allow cross-comparison of their spatial distributions. Earlier studies mapped and annotated the location of traced cells roughly based on outlines of brain structures in serial sections as compared to the standard stereotaxic atlas (*Franklin and Paxinos, 2008*). It is difficult to compare the spatial organizations for different tracing results, let alone comparing results across laboratories since many anatomical structures in the brainstem are poorly defined. For example, the intermediate reticular nucleus (IRt), where many orofacial premotor neurons reside, extends for ~3 mm long in the adult along the anterior-posterior axis in the adult mouse. Thus, simply

annotating premotor neurons that reside in IRt does not pinpoint their exact spatial location. Therefore, mapping premotor neurons in a standard coordinate frame and reconstructing them in the same three-dimensional space with high spatial resolution would greatly facilitate subsequent functional interrogation of orofacial premotor circuits.

To meet the above challenges, we developed a new three-step monosynaptic rabies virus-based strategy to trace orofacial premotor circuits in adult mice (*Figure 1B*). We used this new approach to delineate premotor circuits for vibrissa (whisking), genioglossus (tongue protrusion), and masseter (jaw-closing) motoneurons. The traced premotor neurons were all registered in the Allen Mouse Brain Common Coordinate Framework (CCFv3) (*Wang et al., 2020*). Three-dimensional reconstruction further enabled visualization of the full picture of their relative organization and distributions. The coordinates of all traced premotor neurons are accessible from the source file for interested users.

## Results

### Adult premotor circuit tracing strategy

To achieve monosynaptic orofacial premotor circuit tracing in adult mice, we developed a three-step monosynaptic rabies virus tracing strategy (*Figure 1B*). First, we introduce Cre recombinase into motoneurons innervating specific muscles through the intramuscular injection of the highly efficient retrograde viral vector AAV2-retro-Cre (*Tervo et al., 2016*) in juvenile mice. Second, in adult mice, we inject Cre-dependent AAV that expresses the TVA receptor and the optimized rabies glycoprotein oG (AAV-Flex-TVA-mCherry [*Miyamichi et al., 2013*] and AAV-Flex-oG [*Kim et al., 2016*]) into the corresponding brainstem motor nuclei in adult mice. In this way, TVA-mCherry and oG are specifically expressed in motoneurons that innervate the muscle that previously had AAV2-retro-Cre injection. Finally, we inject the pseudotyped EnvA-ΔG-RV-GFP, which only infect TVA-expressing motoneurons. To further reduce any non-specific background infection by EnvA-ΔG-RV-GFP, we used a mutated version of the envelope, EnvA(M21), for pseudotyping -ΔG-RV. Thus, EnvA(M21)-RV-ΔG-GFP was used for all of our experiments (*Sakurai et al., 2016*). EnvA(M21)-ΔG-RV is also called as CANE-ΔG-RV. Five days later, through the complementation of oG, the virus will spread into the corresponding premotor neurons (*Wickersham et al., 2007*).

To determine the efficiency and specificity of the AAV2-retro-Cre virus transduction of intended motoneurons, we injected the Cre virus into the mystacial pad of different aged mice and examined the expression of Cre-dependent tdTomato, using Ai 14 mice, in the facial motor nucleus (FN), where the myotopic map is well-described (*Figure 1—figure supplement 1*; *Ashwell, 1982*; *Deschênes et al., 2016b*; *Furutani et al., 2004*; *Guest et al., 2018*; *Hinrichsen and Watson, 1984*; *Klein and Rhoades, 1985*; *Komiyama et al., 1984*; *Sreenivasan et al., 2015*; *Terashima et al., 1993*; *Watson et al., 1982*). When we injected AAV2-retro-Cre early in postnatal days (~P10) into the mystacial pad, we observed widespread labeling. In addition to motoneurons located in the lateral part of FN that innervates the mystacial pad, neurons in the medial and middle parts of FN were also labeled, likely through entering the fluid of the peri-nerve space after injection and subsequently infecting other motor axons in the same nerve. As the age of the mouse advanced, the retrogradely labeled neurons became progressively restricted to the lateral part of FN and the number of labeled neurons also drastically decreased (*Figure 1—figure supplement 1*). We, therefore, decided to inject AAV2-retro-Cre in the desired craniofacial muscles at P17 as step 1, which gave us specificity and good efficiency of infection. For step 2, at more than 3 weeks later, helper viruses (AAV-Flex-TVA-mCherry and AAV-Flex-oG) would be injected into the corresponding motor nucleus, and for step 3, 2 weeks after helper AAVs were injected, EnvA(M21)-ΔG-RV-GFP would be injected into the same nucleus (*Figure 1B*). The brains are collected 5 days after RV injection.

We applied the three-step monosynaptic tracing strategy to investigate the premotor circuit for the following motor units: vibrissa motoneurons, with cell bodies in the lateral part of FN; tongue-protruding genioglossus motoneurons of the hypoglossal nucleus; and the jaw-closing masseter motoneurons of the trigeminal motor nucleus. These are hereafter referred to as vibrissa, genioglossus, and masseter premotor circuits, respectively. The tracing results are described in details below. Notably, despite efficient transsynaptic labeling, we often observed a scarce number of TVA-mCherry and GFP double-positive motoneurons, that is, starter cells, and the majority of those neurons degenerated. The locations of the remaining starter cells in the respective motor nuclei in all samples are shown in

*Figure 5—figure supplement 1A,B* (FN), *Figure 5—figure supplement 2A,B* (hypoglossal nucleus), and *Figure 5—figure supplement 3A,B* (trigeminal motor nucleus). This loss of starter cells results from the toxicity of ΔG-RV since omission of RV injection did not cause this problem (data not shown).

## Vibrissa premotor circuit

We first consider a qualitative overview of the labeling. The densest labeling of vibrissa premotor neurons was found in the Bötzinger complex (BötC)/retrofacial region (*Figure 2A*), the vibrissa zone of the IRt (vIRt) (*Figure 2B*), and the dorsal medullary reticular nucleus (MdD) (*Figure 2E*). The BötC/ retrofacial that region resides immediately posterior to FN is known to contain expiration-rhythmic cells (*Deschênes et al., 2016a*) and is implicated in the control of sniffing behavior that is often coupled with whisking during exploration (*Deschênes et al., 2012*). The vIRt, located medial to the compact part of the nucleus ambiguus (cNA), is known to contain whisking oscillator cells. A few premotor neurons were also consistently observed in the preBötzinger complex (preBötC) (*Figure 2B*), which is reported to entrain whisking and sniffing (*Moore et al., 2013*).

We now consider sites of inputs from vibrissa primary afferents, either within or adjacent to the ipsilateral spinal trigeminal nuclei, which receive inputs from vibrissa primary afferents. Those areas include the spinal trigeminal nucleus oralis (SpVO) (*Figure 2C*), rostral part of the interpolaris (SpVIr) (*Figure 2D*), and the muralis (SpVm, data not shown). Rostral to FN, we observed labeling in the ipsilateral Kölliker-Fuse (KF) (*Figure 2F*), the bilateral midbrain reticular formation (MRN, near the red nucleus) (*Figure 2G*), and the superior colliculus (SC) with contralateral dominance (*Figure 2H*). The SC contains two clusters of vibrissa premotor neurons (*Figure 2—figure supplement 1A,B*). The caudal cluster (peak density; anterior-posterior (AP) –3.73 ± 0.16 mm, medial-lateral (ML) 1.34 ± 0.03 mm, dorsal-ventral (DV) –2.57 ± 0.09 mm; four mice) resides in the intermediate layer of SC, whereas the rostral cluster locates in the deep layer of SC (peak density; AP – 3.61 ± 0.09 mm, ML 1.09 ± 0.04 mm, DV –2.57 ± 0.09; four mice).

We strived to maintain normal whisking in animals over the 3-week period of time between injection into juvenile and analysis of adult animal. Thus, we did not lesion any of the nerves that drive the extrinsic muscles, which control the position of the mystacial pad (*Dörfl, 1985*) lest this lead to unanticipated alteration in the adult whisking circuit. Thus, we could not rule out the possibility of infecting a few extrinsic motoneurons much as we aimed our injection at the intrinsic vibrissa muscles that control vibrissa protraction. The distribution of vibrissa premotor neurons observed in adult mice is largely consistent with the pattern observed previously in perinatal tracing experiments (*Takatoh et al., 2013*). However, we did find a few differences between adult and postnatal circuits. First, vibrissa premotor neurons were labeled in the ipsilateral deep cerebellar nucleus (DCN) interpositus (*Figure 2I*) in the adult mouse, which were not present in juvenile animals. Second, we did not find premotor neurons in the spinal vestibular nucleus in adult mice, although this was observed in perinatal tracing. Third, the clusters of premotor cells in the lateral paragigantocellular nucleus (LPGi) bilaterally in neonatal mice became less distinct in adult. The neurons in LPGi might have migrated medially in post-juvenile development, noting that we observed the larger number of labeled cells in the gigantocellular reticular nucleus, which situates medial to LPGi, at the level of the FN in adult mice. Finally, we observed labeled neurons in the zona incerta (ZI) and in extended amygdala in adult that were not labeled in the neonatal transsynaptic tracing studies (*Figure 2—figure supplement 1C,C'; Takatoh et al., 2013*). Collectively, adult premotor tracing revealed a similar spatial distribution pattern in the brainstem reticular and sensory nuclei in juvenile and adult mice with both addition and loss of vibrissa premotor neurons in a few areas.

## Tongue-protruding premotor circuit

Qualitatively, the greatest number of labeled tongue-protruding premotor neurons was found bilaterally in the dorsal IRt (*Figure 3A, B*). These neurons spread along the anterior-posterior axis of the dorsal IRt with the highest density in the area anterior to the rostral edge of the hypoglossal nucleus (see the details of IRt organization in the section below). Extending from the dorsal to ventral IRt (where vIRt resides), labeling gradually became sparser. Lateral and rostral to IRt, we also observed premotor neurons with relatively larger sizes, that is, compared to IRt in the parvicellular reticular nucleus (PCRt) dorsal to the FN, and these cells exhibit medially oriented dendrites (*Figure 3C*). We did not distinguish between the ipsilateral and contralateral results since the left and right hypoglossal

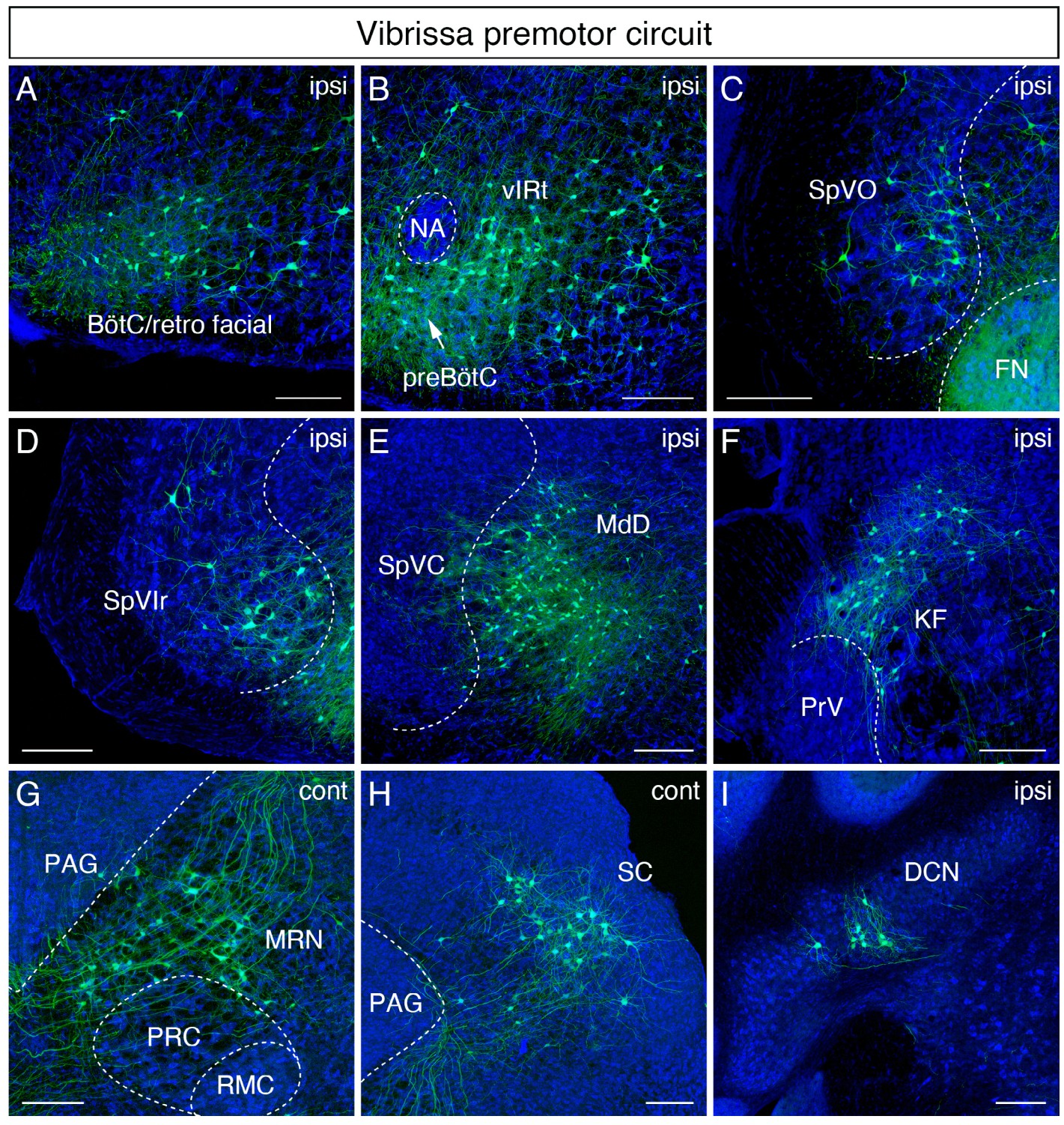

**Figure 2.** Monosynaptic tracing results of vibrissa premotor neurons in adult mice. Representative images of traced vibrissa premotor neurons on coronal sections. Sections were counterstained with fluorescent Nissl (blue). Labeled neurons shown in the ipsilateral Bötzinger complex (BötC)/retrofacial area (**A**), preBötC (arrow) and vibrissal intermediate reticular formation (vIRt) (**B**), spinal trigeminal nucleus oralis (SpVO) (**C**), rostral part of spinal trigeminal nucleus interpolaris (SpVIr) (**D**), medullary reticular nucleus dorsal (MdD) located medial to spinal trigeminal nucleus caudalis (SpVC) (**E**), Kölliker-Fuse (KF) (**F**), contralateral midbrain reticular nucleus (MRN) located dorsal to the red nucleus parvicellular region (RPC) (**G**), contralateral superior colliculus (SC) (**H**), and ipsilateral deep cerebellar nucleus (DCN) (**I**). Scale bars, 200 µm.

The online version of this article includes the following figure supplement(s) for figure 2:

**Figure supplement 1.** Additional vibrissa premotor inputs.

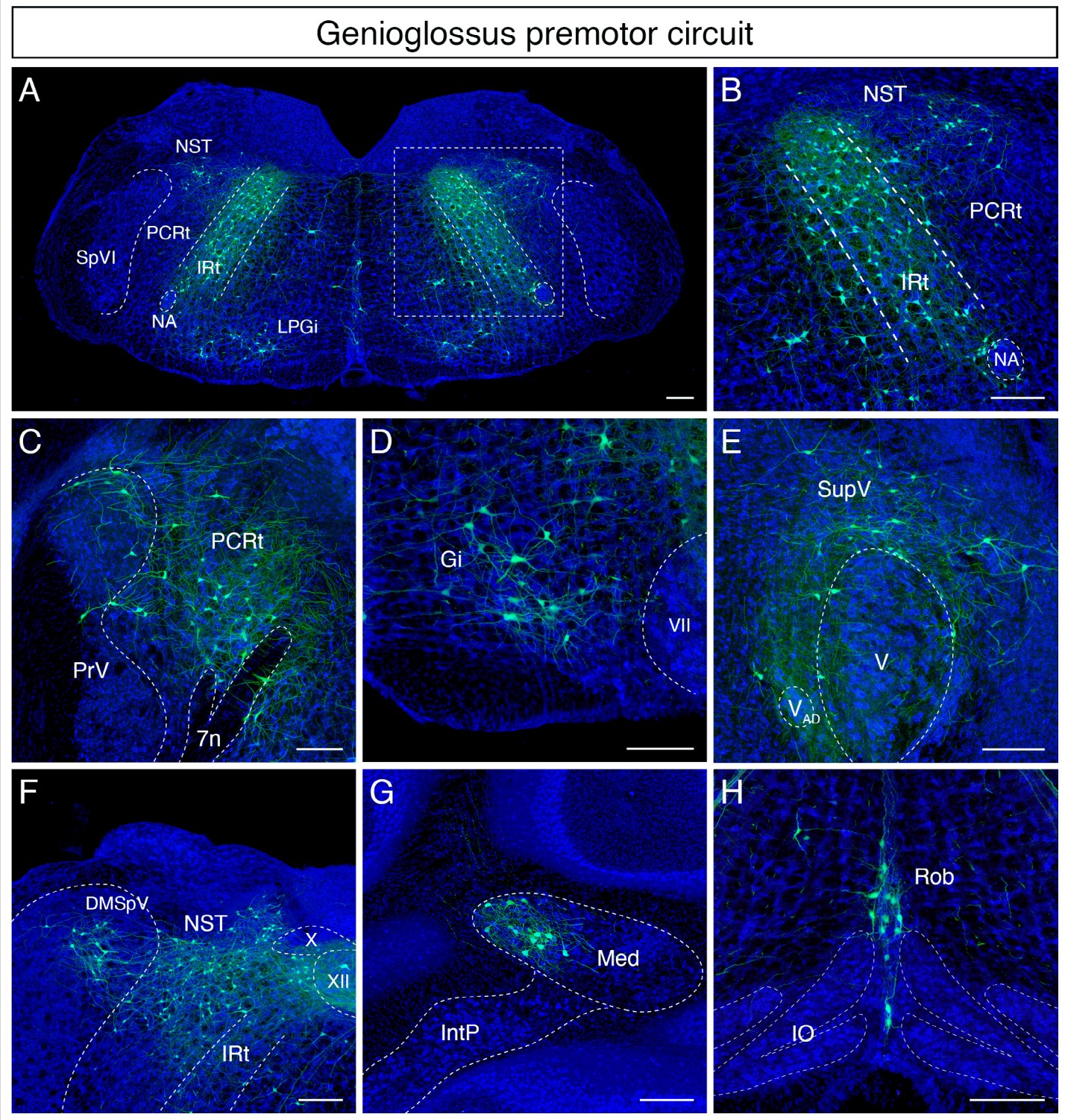

**Figure 3.** Monosynaptic tracing results of tongue-protruding genioglossus premotor neurons in adult mice. Representative images of traced genioglossus premotor neurons on coronal sections. Sections were counterstained with fluorescent Nissl (blue). Labeled neurons are shown in the dorsal intermediate reticular nucleus (IRt), nucleus of solitary tract (NST), lateral paragigantocellular nucleus (LPGi) at the anterior-posterior level between VII and XII (**A**, magnified view of the boxed area in **A** is shown in **B**), parvicellular reticular nucleus (PCRt), dorsal region of the principal trigeminal nucleus (PrV) (**C**), gigantocellular reticular nucleus (Gi) (**D**), supratrigeminal region (SupV) (**E**), dorsomedial part of spinal trigeminal nucleus (DMSpV), rostral NST at the anterior-posterior level of the anterior part of XII (**F**), the medial subnucleus of the deep cerebellar nucleus (DCN) (**G**), and raphe obscurus nucleus (Rob) (**H**). Scale bars, 200 μm.

nucleus are adjacent to the midline, and the genioglossus muscles are located near the midline, such that the AAV2retro virus could infect motoneurons on both sides of the brainstem.

Many labeled cells with very large soma size were found in Gi/LPGi/LRt areas, spanning along the anterior-posterior axis (*Figure 3D*). In the pons, labeled premotor neurons were found in the supratrigeminal nucleus and peritrigeminal zone around the trigeminal motor nucleus (*Figure 3E*). In the sensory-related areas, labeled tongue premotor neurons were observed in the dorsal part of the principal trigeminal nucleus (PrV) (*Figure 3C*), nucleus of solitary tract (NST), and dorsomedial SpV (DMSpV) (*Figure 3F*). In the cerebellum, labeled neurons resided in the medial subnucleus of DCN (*Figure 3G*). Additional premotor input was found in the raphe obscurus nucleus (Ro) (*Figure 3H*). The distribution of the adult genioglossus premotor neurons described above is similar to the pattern observed in juvenile mice (P8 > P15 transsynaptic tracing) (*Stanek et al., 2014*). However, in adult mice, the large cluster of premotor neurons in the dorsal midbrain reticular formation (dMRf) previously found in juvenile animals (*Stanek et al., 2014*) was absent in adult mice.

## Jaw-closing premotor circuit

Qualitatively, in the masseter premotor circuit, extensive labeling was also found bilaterally along the anterior-posterior axis of the dorsal IRt (*Figure 4A and B*). Interestingly, the majority of labeled dorsal IRt neurons were observed *contralaterally* in the caudal part of IRt (*Figure 4C*). Bilateral labeling in PCRt was observed as a lateral continuum of the dorsal IRt neurons at the level of the FN (*Figure 4B*). Rostrally, we found a distinct bilateral cluster of large-size neurons with medially directed dendrites situated around the PCRt/PrV area immediately caudal to the trigeminal motor nucleus (*Figure 4D*). This group of neurons wedged into the dorsomedial and ventrolateral PrV. This area, identified by Nissl staining, contains a distinct cluster of neurons with large size than neighboring cells. Similar to the tongue-protruding circuit but with fewer numbers, cells of very large soma size were labeled ipsilaterally along the anterior-posterior axis spanning Gi/LPGi/LRt areas (*Figure 4A*). In the pons, numerous labeled masseter premotor neurons were also observed in the supratrigeminal nucleus and peritrigeminal areas (*Figure 4E*).

In the sensory-related areas, labeled cells resided bilaterally in the dorsal PrV (*Figure 4E*), ipsilaterally in the dorsomedial SpV (*Figure 4A*), and ipsilaterally in the mesencephalic trigeminal nucleus (*Figure 4F*). Lastly, within the cerebellum, we identified labeled neurons in the contralateral medial subnucleus of DCN (*Figure 4G*). The distribution of jaw-closing premotor neurons in the adult is similar to the pattern observed in juvenile mice (P8 > P15 transsynaptic tracing) (*Stanek et al., 2014*). However, as with the tongue premotor circuit, cells in dMRf observed in juvenile mice are absent in the adult circuit.

## Mapping orofacial premotor neurons onto Allen common coordinate framework (CCF)

To generate a standardized orofacial premotor atlas that enables cross-comparison of different premotor circuits, RV-traced GFP-positive premotor neurons were mapped onto the Allen Mouse Brain Common Coordinate Framework (CCFv3) (*Wang et al., 2020*). The CCF is a widely used open-access three-dimensional standardized brain atlas generated from the average of 1675 adult C57BL/6J mice. Registration of labeled neurons in CCF enables direct comparison of the results from different laboratories in the same coordinate space. Locations of RV-traced premotor neurons were translated into CCF coordinates using a method modification of a previously described method with our modifications (*Shamash et al., 2018*). In brief, each coronal section was registered to a corresponding CCF plane through a diffeomorphic transformation (for details, see Materials and methods). Subsequently, the labeled cells were identified and counted semiautomatically or manually, and their coordinates were also transformed into CCF coordinates (*Figure 5A*). All traced orofacial motor neurons for vibrissa (four mice), genioglossus (four mice), and masseter (four mice) were registered to the CCF, and their coordinates are accessible from the source file. The cells in CCF coordinates were reconstructed in two- and three-dimensional spaces using Brainrender (*Claudi et al., 2021*).

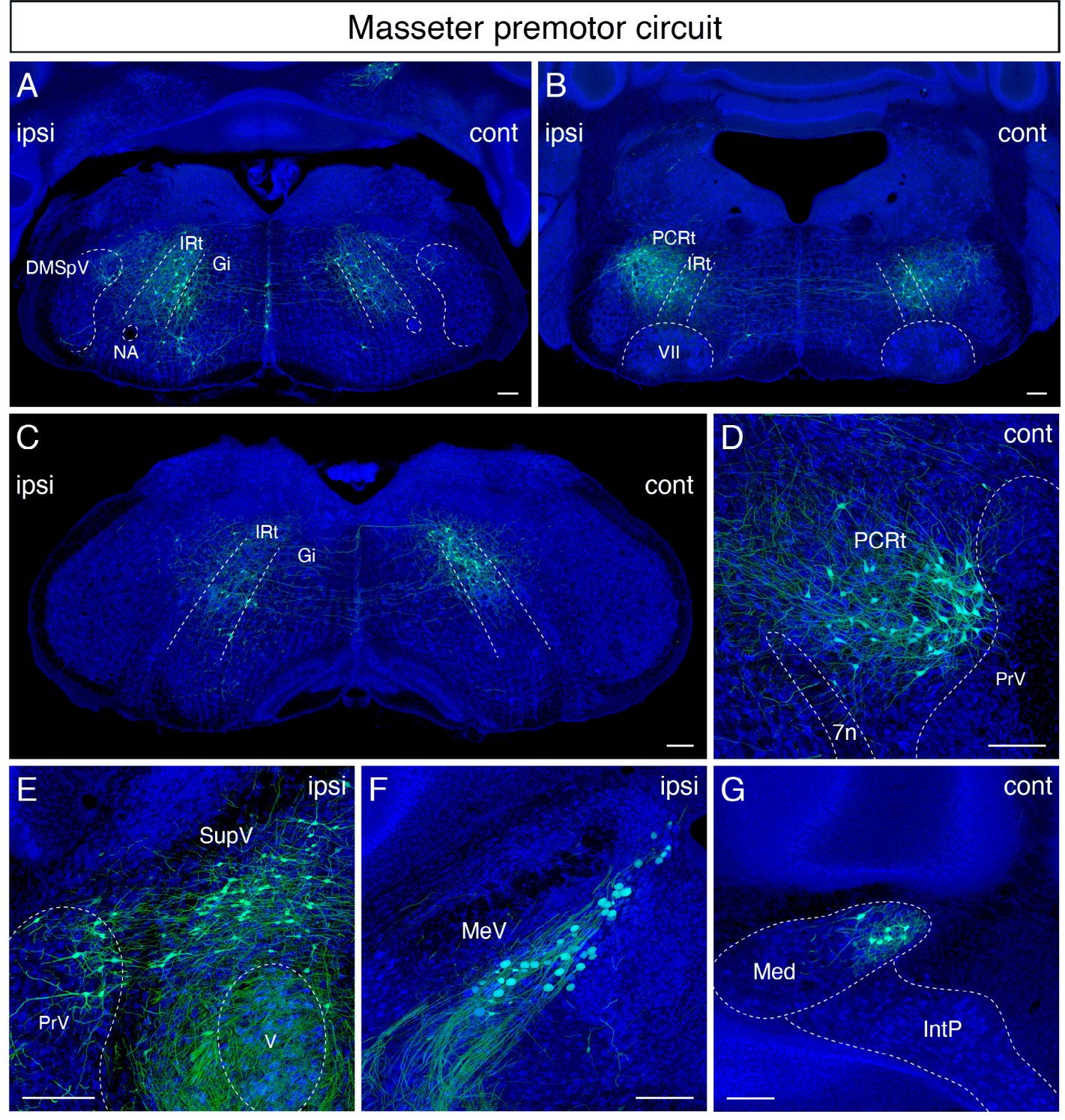

**Figure 4.** Monosynaptic tracing results of jaw-closing masseter premotor neurons in adult mice. Representative images of traced masseter premotor neurons on coronal sections. Sections were counterstained with fluorescent Nissl (blue). Labeled neurons are observed bilaterally in the dorsal intermediate reticular nucleus (dorsal IRt), in dorsomedial part of spinal trigeminal nucleus (DMSpV) at the anterior-posterior level between VII and XII (**A**), bilaterally in the dorsal IRt, parvicellular reticular nucleus (PCRt) at the anterior-posterior level of VII (**B**), contralaterally in the dorsal IRt at the anterior-posterior level of the anterior part of XII (**C**), PCRt (**D**), supratrigeminal region (SupV), dorsal principal trigeminal nucleus (PrV) (**E**), ipsilateral mesencephalic nucleus (MeV) (**F**), PCRt, dorsal PrV (**C**), gigantocellular reticular nucleus (Gi) (**D**), SupV (**E**), and the contralateral medial subnucleus of deep cerebellar nucleus (DCN) (**G**). Scale bars, 200 μm.

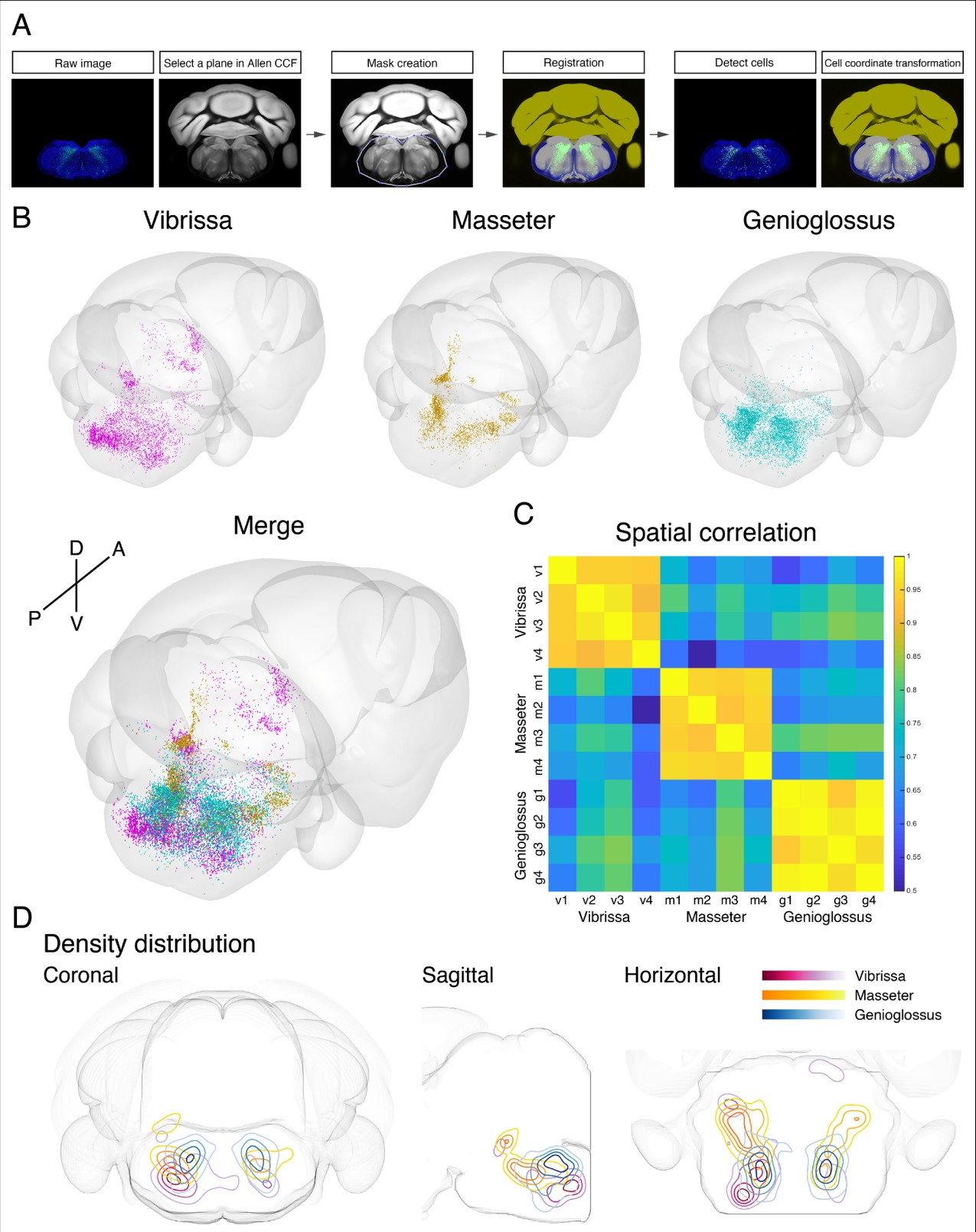

**Figure 5.** Co-registration and comparison of the spatial distributions of vibrissa, genioglossus, and masseter premotor circuits in Allen common coordinate framework (CCF). (**A**) Procedure for mapping orofacial premotor neurons to Allen CCF. (**B**) Reconstructed representative vibrissa (magenta), masseter (gold), and genioglossus (cyan) premotor circuits in Allen CCF (top). Merged image (bottom). (**C**) Cross-correlation analysis of the spatial distribution patterns of individual animals; vibrissa (v1–v4; four mice), masseter (m1–m4; four mice), and genioglossus (g1–g4; four mice) premotor

*Figure 5 continued*

circuits. (**D**) Two-dimensional contour density analysis of representative vibrissa (magenta), masseter (yellow), and genioglossus (blue) premotor circuits.

The online version of this article includes the following video and figure supplement(s) for figure 5:

**Source data 1** Coordinates of all labeled premotor neurons.

**Source data 2** Coordinates of all starter cells.

**Figure supplement 1.** Distribution of starter cells in the facial motor nucleus and labeled vibrissa premotor neurons from individual animals in Allen mouse brain common coordinate framework.

**Figure supplement 2.** Distribution of starter cells in the hypoglossal nucleus and labeled genioglossus premotor neurons from individual animals in Allen mouse brain common coordinate framework.

**Figure supplement 3.** Distribution of starter cells in the trigeminal motor nucleus and labeled masseter premotor neurons from individual animals in Allen mouse brain common coordinate framework.

**Figure supplement 4.** Vibrissa, genioglossus, and masseter premotor neurons overlapped with respect to the Allen mouse brain common coordinate framework.

**Figure supplement 5.** Quantification of transsynaptically labeled neurons in top 10 labeled brain areas for each motor group based on Allen mouse brain common coordinate framework (CCF) nomenclature.

**Figure 5—video 1.** Three-dimensional reconstructed vibrissa (magenta), genioglossus (cyan), and masseter (gold) premotor circuits.

https://elifesciences.org/articles/67291/figures#fig5video1

## Cross-comparison of spatial distributions of vibrissa, genioglossus, and masseter premotor circuits

To compare the spatial organization of vibrissa, genioglossus, and masseter premotor circuits, trans-synaptic labeling results from individual animals were reconstructed in the same CCF space (*Figure 5B and Figure 5—video 1*). Reconstructed premotor circuits for each of the target muscle/motoneurons from different animals are shown in *Figure 5—figure supplements 1–3*. Using the extracted spatial coordinates of all labeled neurons, we performed cross-correlation analysis of the spatial distribution patterns of tracing results from all samples (see Materials and methods). Individual premotor tracing results from the same muscle/motor unit were highly correlated, whereas results obtained from different muscles/motor units showed low correlations in spatial patterns (*Figure 5C*, *Figure 5—figure supplements 1–3*).

Plots of the premotor neurons for vibrissa/genioglossus/masseter into the same CCF in three-dimensions revealed both overlapping and segregating features of these different premotor circuits (*Figure 5—figure supplement 4*, *Figure 5—video 1*). The density plots for each premotor circuit also support the muscle-specific differential spatial organizations, as shown for across anatomical planes (*Figure 5D*). All three premotor circuits showed the highest density of labeling in the intermediate and parvicellular reticular formations (IRt and PCRt); however, the exact peak density positions were in the different locations of IRt/PCRt for different circuits. The vibrissa premotor circuit showed highest labeling density in the caudoventral areas of IRt (*Figure 5D*, red). The masseter premotor circuit had densest labeling in the anterodorsal area of IRt (*Figure 5D*, yellow). The highest density area of the genioglossus premotor neurons is located in IRt between the peaks for the vibrissa and masseter premotor cells (along the A–P axis), although there were shared regions between genioglossus and masseter premotor distributions (*Figure 5D*, blue). Finally, the extracted coordinates for each of the labeled cells enabled automatic assignment of their corresponding anatomical structure used by the CCF. We thus can readily obtain the top 10 transsynaptically labeled premotor nuclei for each muscle recognized by the CCF (*Figure 5—figure supplement 5*). These analyses have collectively provided an overview of the differential anatomical organizations of vibrissa, genioglossus, and masseter premotor circuits in adult mice.

## Detailed comparison of spatial organization orofacial premotor circuits within IRt

Our adult tracing results indicate IRt as the common area of premotor neurons for all three circuits. Earlier studies have either localized or implicated IRt as the region containing oscillator neurons for several orofacial actions, that is, whisking, licking, and chewing rhythm. Yet IRt is an ill-defined area. We thus examined the relative spatial organizations of different premotor circuits within IRt in greater

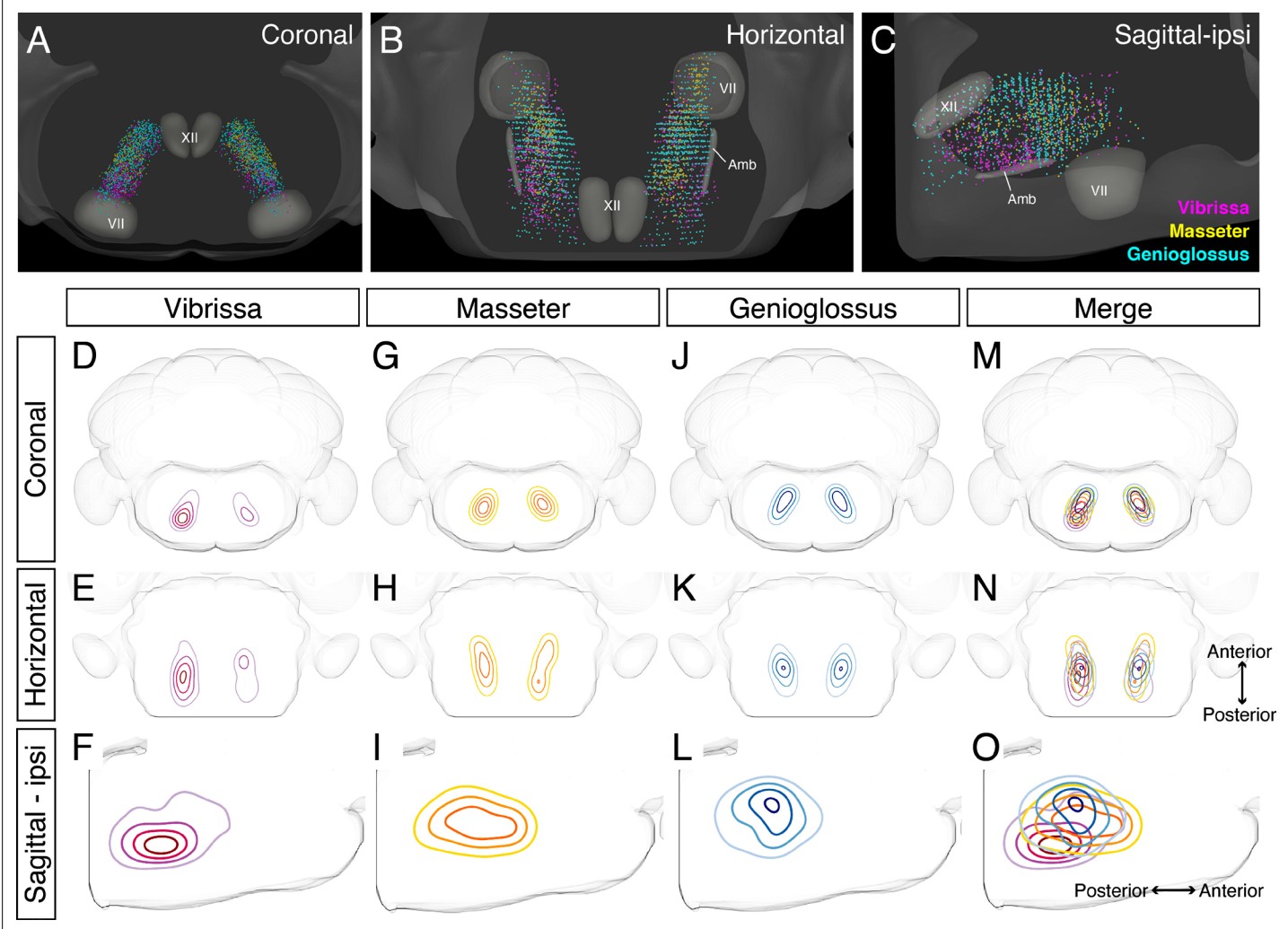

**Figure 6.** Detailed comparison of spatial organizations of orofacial premotor circuits within intermediatereticular nucleus (IRt). (**A–C**) Distribution of vibrissa (magenta), masseter (gold), and genioglossus (cyan) premotor neurons within IRt from representative animals in coronal (**A**), horizontal (**B**), and sagittal (**C**) planes. (**D–O**) Density analysis of vibrissa (**D–F**, magenta, an average of four mice), masseter (**G–I**, yellow, an average of four mice), genioglossus (**J–L**, blue, an average of four mice) premotor neuron distributions. Merged images (**M–O**).

details by taking advantage of our reconstructions of vibrissa, genioglossus, and masseter IRt premotor neurons in the same CCF space to demarcate only cells within IRt (*Figure 6*, locations of craniofacial motor nuclei were also shown as landmarks). These three-dimensional reconstructions revealed partial overlapping and partial segregation of the three premotor circuits (*Figure 6A-C*). Along the A–P axis, the highest density regions of ipsilateral jaw-closing and tongue-protruding premotor neurons in IRt were close to each other, but with the peak of jaw premotor neurons shifted rostrally and ventrally (*Figure 6G-O* , jaw peak: AP –6.02 ± 0.18 mm, DV –6.45 ± 0.04 mm; four mice, tongue peak: AP –6.20 ± 0.23 mm, DV –5.74 ± 0.29; four mice). Along the D–V axis, while tongue premotor neurons are concentrated to more dorsal IRt than jaw premotor neurons, their distribution spread more to ventral IRt. Notably, the contralateral jaw IRt premotor neurons formed a discernible cluster caudal to the densest area of tongue IRt premotor neurons, displaying a bilaterally asymmetric distribution (*Figure 6H*). Vibrissa premotor neurons were more spatially separated from tongue and jaw premotor neurons in IRt, that is, at more caudal and ventral (AP –6.45 ± 0.19 mm, DV –6.45 ± 0.04 mm; four mice) locations in IRt (*Figure 6D-F*). Furthermore, the tongue and jaw IRt premotor neurons showed similar densities between the ipsilateral and contralateral side, which might be expected as licking and chewing generally involve muscles of both sides. In contrast, the vibrissa IRt premotor neurons showed biased distribution to the ipsilateral side (*Figure 6D, E*). Collectively, these results suggest

that functionally distinct groups of orofacial premotor neurons occupy the overlapping yet distinct spatial positions within IRt. Further, there is roughly a ventral-to-dorsal and caudal-to-rostral gradient of vibrissa-tongue-jaw premotor neurons.

## Axon collaterals revealed common premotor neurons for distinct motor neurons

Orofacial behaviors often require coordinated activity of multiple pools of motoneurons. A premotor neuron that simultaneously innervates distinct motoneurons forms the simplest motor coordinating circuit. We therefore examined whether our premotor tracing results provide evidence for the existence of such common premotor neurons. Bright fluorescent signal from RV traced cells allows us to visualize their axons and collaterals. Interestingly, in genioglossus premotor tracing studies, axonal collaterals from a fraction of labeled premotor neurons were found in the middle part of the FN, $VII_{middle}$ (*Figure 7B*), where motor neurons controlling lip and jaw (platysma) movements reside. Axon collaterals were also found densely innervating the small subnucleus of the trigeminal motor nucleus, $V_{AD}$ (*Figure 7C and C'*), which controls the jaw-opening anterior digastric muscles, as well as were observed in the accessory FN (data not shown), which innervates the posterior digastric jaw-opening muscle. These results suggest that certain premotor neurons that control tongue protrusion also simultaneously control mouth- and jaw opening through their axon collaterals, providing a neuronal substrate for coordinating multiple motor groups needed for proper execution of behaviors such as licking and feeding.

Similar to the above case for tongue-tracing studies, in masseter premotor tracing studies, the axonal collaterals from a fraction of labeled jaw-closing premotor neurons were observed in $VII_{middle}$ (*Figure 7D*) (same region receiving innervations from the tongue premotor neurons), in the contralateral trigeminal motor nucleus (*Figure 7E*), and densely in the dorsal part of the hypoglossal nucleus (*Figure 7F*), where motor neurons for tongue retrusion reside. These data provide additional evidence that premotor neurons controlling jaw-closing muscle also simultaneously modulate the tongue retrusion and likely mouth closing muscles through their axon collaterals. In this manner, behaviorally synergistic motor units are coordinately activated to enable proper actions, that is, chewing without biting into the tongue. Finally, we also observed that collaterals from vibrissa premotor neurons project to the contralateral later FN where vibrissa motoneurons reside (*Figure 7A*), but not to hypoglossal and trigeminal motor nuclei. The former projections likely coordinate bilateral control of vibrissa set-point (*Kleinfeld et al., 2014*).

Where might the common premotor neurons that send collaterals to multiple brainstem motor nuclei reside? Dense axon collaterals projecting to $V_{AD}$ and $VII_{middle}$ motoneurons from genioglossus premotor neurons inspired us to use a retrograde split-Cre strategy to trace common premotor neurons that innervate both XII (where motoneurons for genioglossus reside) and $VII_{middle}$. In this strategy, functionally inactive halves of Cre (CreN and CreC) packaged in retrograde lentivirus (RG-LV [*Kato et al., 2011*]; RG-LV-CreN, RG-LV-CreC) were separately injected into $VII_{middle}$ and XII of Ai14 mice (*Figure 7G*; *Stanek et al., 2016*; *Wang et al., 2012*). In this injection scheme, functional Cre is reconstituted, and tdTomato is visualized only in neurons simultaneously innervating $VII_{middle}$ and XII (*Figure 7H*). Retrograde split-Cre tracing revealed tdTomato-positive cells in SupV and the dorsal IRt areas (four mice, *Figure 7K and L*), where genioglossus premotor neurons were found (*Figure 3A, B and E* and *Figure 5—figure supplement 2D*). Notably, in addition to $VII_{middle}$ and XII, we found tdTomato-positive axon terminals onto motoneurons in $V_{AD}$ (jaw opening) and in the nucleus ambiguus (mostly in semi-compact part), which are known to be involved in swallowing. Thus, $VII_{middle}$-XII common premotor neurons located in SupV and dorsal IRt simultaneously innervate motoneurons controlling tongue protrusion, lower lip, jaw opening, and throat (through the nucleus ambiguus) (*Figure 7O and P*), suggesting that those common premotor neurons likely represent a fundamental neural substrate for coactivating these muscles. Interestingly, SupV and dorsal IRt were also labeled by the retrograde split-Cre tracing from the left and right sides of $VII_{middle}$ (*Figure 7—figure supplement 1*). Those neurons also project additional collaterals to V and XII, in addition to $VII_{middle}$. These results indicate that SupV and dorsal IRt regions may be critical brainstem hubs that contain common premotor neurons that coordinate multiple groups of motoneurons for orofacial feeding behaviors (*Figure 7P*).

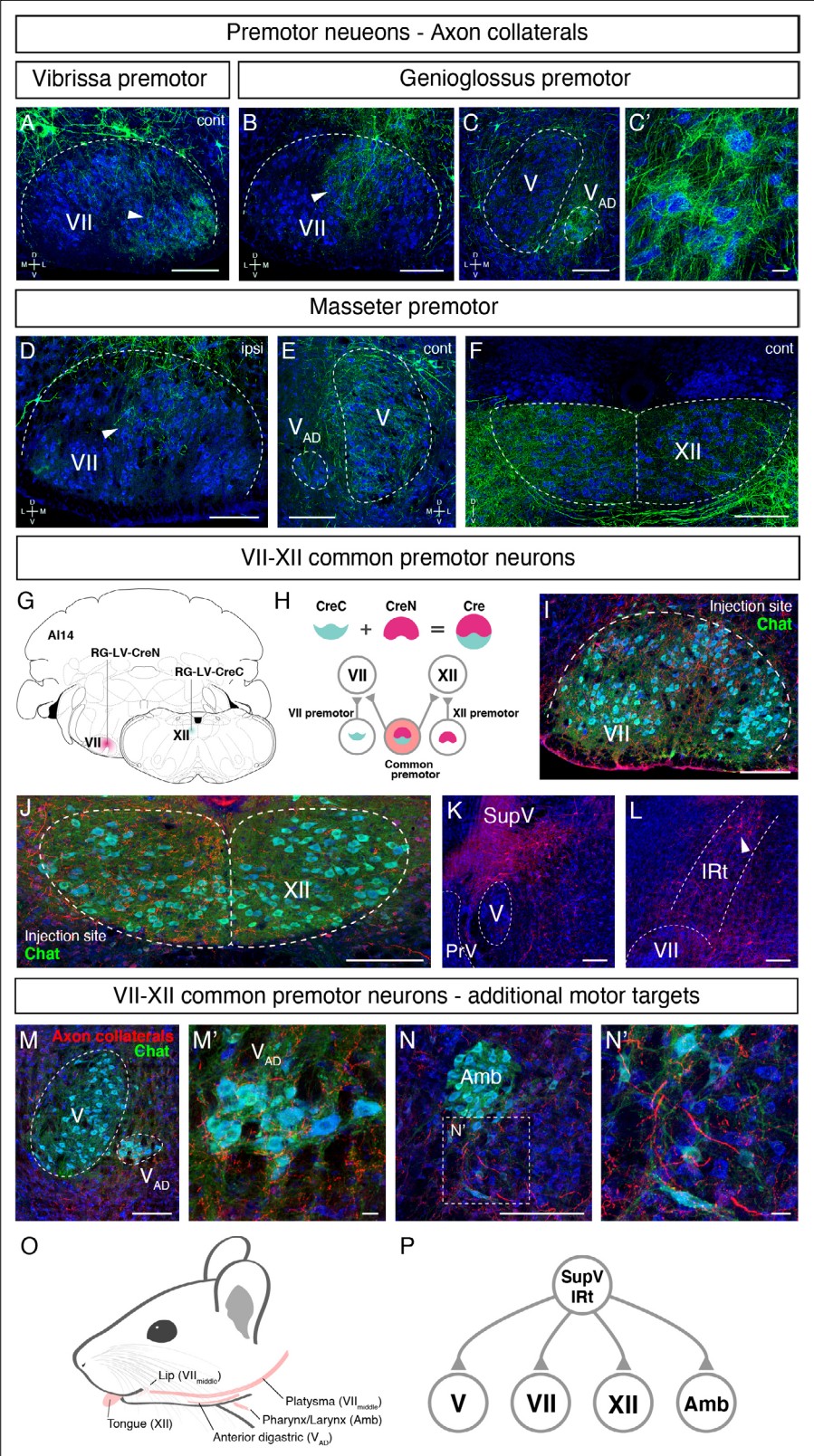

**Figure 7.** Common premotor neurons innervate multiple distinct orofacial motor nuclei. (**A–F**) Representative images of axon collaterals from rabies labeled premotor neurons traced from one muscle innervating other orofacial motor nuclei. Sections were counterstained with fluorescent Nissl (blue). (**A**) Axon collaterals from *ipsilateral* vibrissa premotor neurons innervate the *contralateral* vibrissa motoneurons in the lateral part of VII

*Figure 7 continued on next page*

*Figure 7 continued*

(arrowhead). (**B–C'**) Axon collaterals of some genioglossus premotor neurons also innervate the middle part of VII$_{middle}$ (arrowhead, **B**) and innervate the anterior digastric part of V (V$_{AD}$) (**C**, magnified view is shown in **C'**). (**D–F**) Axon collaterals from masseter premotor neurons also innervate the middle part of VII$_{middle}$ (**D**), the contralateral V (**E**), and the dorsal part of XII (**F**). (**G–P**) Identifying VII$_{middle}$-XII common premotor neurons. (**G, H**) Schematic of split-Cre tracing strategy. (**G**) RG-LV-CreN and RG-LV-CreC were injected into the left side of VII$_{middle}$ and XII of Ai 14 mice, respectively. (**H**) Cre is reconstituted only in neurons innervating both VII$_{middle}$ and XII, and which induces tdTomato reporter expression. (**I, J**) Representative images of axons/axon collaterals in the injection sites. Sections were counterstained with fluorescent Nissl (blue). Motoneurons were stained with anti-chat antibody (green). VII (**I**). XII (**J**). (**K, L**) Representative images of VII$_{middle}$-XII common premotor neurons in supratrigeminal region (SupV) (**K**) and the dorsal intermediatereticular nucleus (IRt) (**L**). (**M–N'**) Representative images of axon collaterals from VII$_{middle}$-XII common premotor neurons in V$_{AD}$ (**M**, magnified view is shown in **M'**) and Amb (**N**, magnified view is shown in **N'**). Scale bars, 200 µm (**A–F, I–N**); 20 µm (**C', M', N'**). (**O**) Schematic showing orofacial muscle targets of motor nuclei. (**P**) Schematic of all motor nuclei innervated by VII$_{middle}$-XII common premotor neuron in SupV and IRt.

The online version of this article includes the following figure supplement(s) for figure 7:

**Figure supplement 1.** Identifying common premotor neurons with bilateral collateral projections to VII$_{middle}$.

## Discussion

We developed a three-step monosynaptic RV tracing to trace the premotor circuits for three different orofacial muscles in adult mice (*Figure 8*). We registered and reconstructed all the traced neurons in the standardized Allen mouse CCF and consequently generated the atlas showing positions of different orofacial premotor circuits in a common brain. This common atlas uncovers the overlapping yet distinct spatial organizations of premotor neurons involved in controlling movements of vibrissae, the tongue, and the jaw. Visualization of premotor neurons' axon collaterals and retrograde split-Cre tracing studies further highlighted premotor neurons in SupV and dorsal IRt as potential substrates for coordinating multiple distinct orofacial muscles involved in feeding-related behaviors. Since these three groups of motoneurons are involved in three rhythmic orofacial behaviors, whisking, licking, or chewing, we next focus our discussion on the implications of the premotor atlas for rhythm generation.

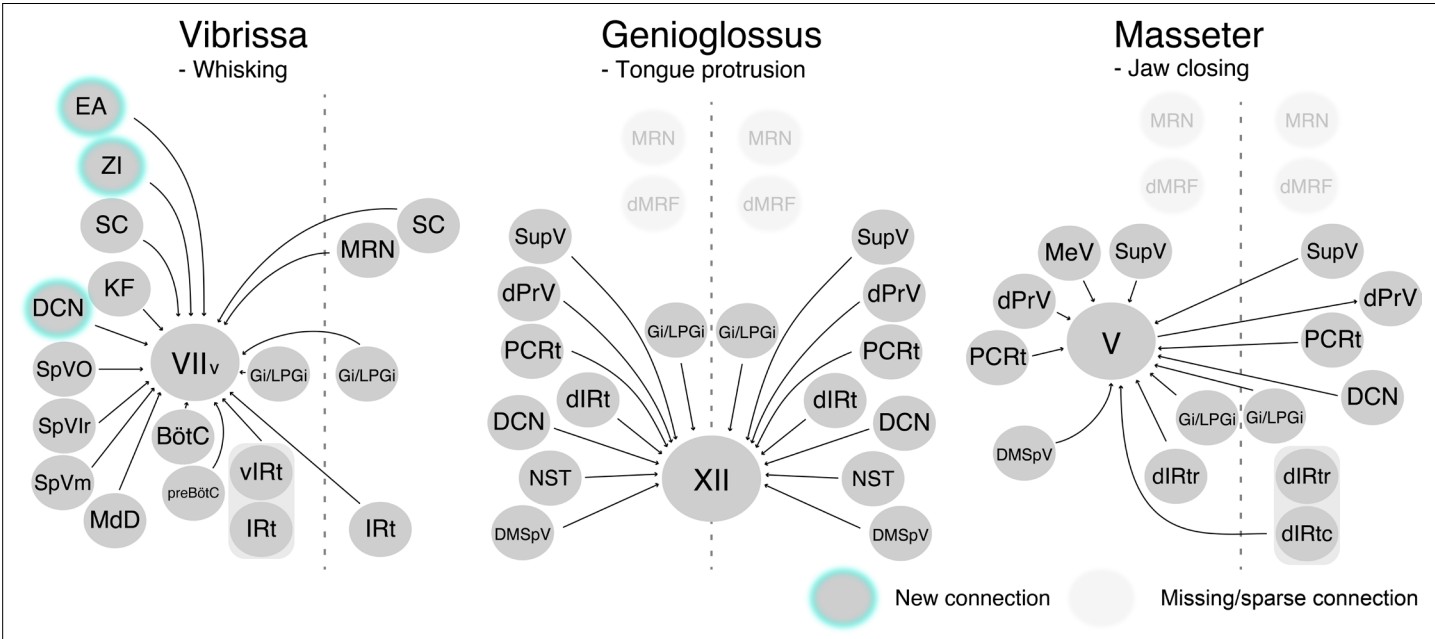

**Figure 8.** Schematic of vibrissa, tongue-protruding genioglossus, and jaw-closing premotor circuits in the adult mice. Newly emerged connections in adults that were not observed in neonates are outlined in turquoise. Neonatal connections that appear lost or becoming sparse are shown as translucent spheres.

## Implications for premotor neurons modulating whisking rhythm

Among the three orofacial premotor circuits in adult mice that we have mapped, the vibrissa premotor atlas consists of the most numerous brain structures (*Figure 8*). This is not surprising since vibrissa movements, as opposed to more stereotyped licking and chewing, are needed for tactile exploration of complex physical environments (*Mehta and Kleinfeld, 2004*; *Prescott et al., 2011*; *Sofroniew and Svoboda, 2015*). Two previous studies uncovered vIRt as the region containing vibrissa oscillator neurons (*Deschênes et al., 2016b*; *Moore et al., 2013*). Indeed, we found extensive labeling of vibrissa premotor neurons in vIRt, with qualitatively more labeled neurons than neonatal tracing. Previous studies also revealed the coupling between breathing/sniffing and whisking (*Moore et al., 2013*; *Welker, 1964*). Along this line, we traced premotor cells in two brainstem areas known to control the respiratory rhythm, that is, the retrofacial/BötC and preBötC (in both juvenile and adult mice), suggesting their roles in coordinating breathing and whisking, and potentially resetting the whisking rhythm (*Kleinfeld et al., 2014*). However, there is an unresolved issue with regard to the inputs from the inspiratory rhythm generator preBötC. PreBötC innervates vIRt; therefore, preBötC is likely pre-premotor for vibrissa motoneurons (*Moore et al., 2013*). Further, tracing studies that used a small injection of sindbis-GFP virus in electrophysiologically identified lateral portion of preBötC revealed that these labeled preBötC neurons project specifically to the lateral and dorsolateral parts of FN, where motoneurons for the nares dilation and extrinsic mystacial pad retraction reside, but rarely to the part where intrinsic vibrissa motoneurons are located (*Deschênes et al., 2016b*). These projections are likely to extrinsic motoneurons that control the position of the mystacial pad (*Dörfl, 1985*), which tracks breathing when whisking dissociates from breathing (*Moore et al., 2013*). In contrast, somatostatin- and glycine-positive preBötC neurons both project to the entire lateral part of FN, including ventral lateral FN where vibrissa-protracting intrinsic motoneurons reside (*Yang and Feldman, 2018*). In our tracing strategy, we injected AAV2-retro-Cre into the mystacial pad in P17 mice; therefore, it is possible that we traced premotor neurons both for intrinsic and extrinsic motoneurons. Further study will be required to understand the precise connection between retrofacial/BötC and preBötC premotor neurons and extrinsic versus intrinsic motoneurons for whisking.

## Implications for premotor neurons modulating licking rhythm

In the tongue-protruding premotor circuit, dorsal IRt near the rostral end of the hypoglossal nucleus contains the highest density of premotor neurons. This area has previously been implicated as the rhythm generator for licking. Using extracellular recording in awake rats, Travers et al. demonstrated that neurons in this area show rhythmic activity that is phase-locked to licking (*Travers et al., 2000*). Furthermore, premotor neurons in this area express cFos after gaping behavior that involves extensive tongue movement (*DiNardo and Travers, 1997*). However, bilateral infusion of muscimol in this area reduces licking electromyogram amplitude with minimal effect on the licking frequency (*Chen et al., 2001*; *Travers et al., 2010*), raising the possibility that dorsal IRt cells are the output of the actual licking oscillator. Importantly, bilateral infusion of muscimol in the same IRt area also suppresses chewing/mastication, indicating the function of the dorsal IRt for coordination of tongue and jaw during ingestion behaviors. We also traced premotor neurons in NST and the adjacent dorsolateral IRt. Interestingly, bilateral infusion of μ-opioid receptor agonist, DAMGO, in the dorsolateral IRt/rostral NST reduces licking frequency with increased amplitude (*Kinzeler and Travers, 2011*). Future studies using premotor neuron-specific manipulations will be necessary to dissect the function of the dorsal IRt and rostral NST for modulating licking and chewing rhythmic motor actions.

## Implications for premotor neurons modulating chewing rhythm

In the jaw-closing premotor circuit, two regions, the dorsal IRt and PCRt between the rostral extent of the FN and the rostral extent of the hypoglossal nucleus, contain the highest numbers of premotor cells. This IRt/PCRt area has been implicated as a critical node for generating chewing rhythm (*Chandler et al., 1990*; *Nakamura et al., 2017*; *Nozaki et al., 1986b*; *Nozaki et al., 1986a*; *Travers et al., 2010*). Nozaki et al. used a fictive rhythmic chewing preparation in guinea pigs to show that stimulation of the cortical masticatory area induces rhythmic activity in what was at the time called the oral part of the gigantocellular (Gi) reticular nucleus (Go). The rhythmic activity is, in turn, conveyed to the trigeminal motor nucleus through premotor neurons in the PCRt (*Nozaki et al., 1986b*; *Nozaki et al., 1986a*). More recently, Travers et al. demonstrated in awake rats that muscimol infusion in rostral

IRt/PCRt but not Go suppresses neuropeptide Y-induced chewing behavior (*Travers et al., 2010*). Nakamura et al. used awake mice to show that infusion of bicuculline in the same IRt/PCRt area evokes chewing (*Nakamura et al., 2017*). Future studies with causal manipulation of IRt/PCRt jaw premotor neurons should provide more definitive answers as to which neurons in IRt/PCRt and how are they involved in generating masticatory rhythms. Since there are also tongue premotor neurons labeled in this area, it will be interesting to know whether these cells also innervate tongue-muscles and coordinating jaw-tongue movements during breaking down of food.

Other studies showed that neurons in the dorsal PrV generate rhythmic bursting in brainstem slices spontaneously and upon electrical stimulation of the trigeminal tract (*Morquette et al., 2015*; *Sandler et al., 1998*), and during fictive chewing in anaesthetized and paralyzed rabbits (*Tsuboi et al., 2003*), suggesting the role of this area for chewing rhythm generation. We also observed jaw premotor neurons in the lateral edge of PCRt, which could be part of SpVO or PrV, with an A–P location at the caudal edge of the trigeminal motor nucleus. Several studies assigned this area as a part of SpVO based on its receptive field in the oral areas (*Inoue et al., 1992*; *Westberg et al., 1995*; *Yoshida et al., 1994*). Neurons in this area in cats respond to either noxious stimulation of the tongue or to light mechanical stimulation of intra- or perioral structures, including the teeth, gingiva, and lip. These neurons issue collaterals that terminate in the trigeminal motor nucleus (*Yoshida et al., 1994*). Inoue et al. demonstrated in rats that premotor neurons in SpVO and the supratrigeminal nucleus share the same masticatory rhythm during cortically induced fictive mastication (*Inoue et al., 1992*). Some of these neurons are activated at short latencies by the stimulation of cortical masticatory area, or stimulation of the inferior alveolar and infraorbital nerves innervating oral areas, or passive jaw opening. Based on these properties, it is suggested that premotor neurons in SpVO integrate sensory information from the oral area and rhythmic activity generated by a central rhythm generator to produce appropriate activity patterns during mastication.

## Common premotor neurons as potential neuronal substrate for coordinating orofacial behaviors

Tracing of axon collaterals from ΔG-RV-GFP-labeled orofacial premotor neurons and the retrograde split-Cre-mediated tracing studies uncovered common premotor neurons with extensive axon collateral network to jaw (trigeminal motor), tongue (hypoglossus), lip-jaw (facial), and throat (nucleus ambiguus), but importantly not to the vibrissae. It should be noted that the retrograde lentivirus used here does not have the motoneuron specificity as the monosynaptic rabies virus. Therefore, the retrograde split-Cre results should be interpreted together with monosynaptic RV tracing results as a secondary verification method. Here, this method revealed SupV and IRt as the major sources of common premotor areas, and importantly, these areas were also identified as the genioglossus and masseter premotor areas by monosynaptic RV tracing. Notably, SupV and IRt also contain bilaterally projecting masseter premotor neurons (*Stanek et al., 2016*). As noted above, neurons in both dorsal IRt and SupV are known to show rhythmic firing during licking and chewing (*Inoue et al., 1992*). Common premotor neurons in SupV and dorsal IRt, therefore, may serve the simplest form of neuronal substrate for coordinating feeding-related orofacial behaviors by means of broadcasting rhythmic information of licking and chewing to synergistic muscles. Future functional manipulation studies are needed to determine their causal functions in coordination of different orofacial behaviors.

## Other notable implications of the adult orofacial atlas

The three-step monosynaptic RV tracing allows us to reveal adult premotor circuits for a specific group of motor neurons, thereby advancing the transsynaptic premotor maps previously only available for neonatal mice. The coordinates of all traced neurons, including starter motoneurons, registered to the Allen mouse brain CCF are accessible from the source file and can be used in the future to guide placement of electrodes for in vivo recordings as mice perform different orofacial behaviors. Comparing to neonatal circuits, we observed new additions of presynaptic inputs to control vibrissa motoneurons from ZI, DCN, and extended amygdala, and loss of presynaptic inputs from dMRf to tongue and jaw motoneurons in the adult premotor circuits (*Figure 8*). These changes may reflect more fine-tuned control of tactile sensation by the vibrissae, and changes in feeding behaviors from neonatal suckling to adult licking and chewing.

One potential alternative interpretation of the difference in the labeling patterns between neonatal and adult mice could be viral neurotropism. We could not rule out the possibility that certain premotor neurons are differentially susceptible to rabies virus infection in neonates and adults. On the other hand, previous classic retrograde tracer injection in the hypoglossal or trigeminal motor nucleus did not identify MRN and dMRf as premotor areas in the adult (*Borke et al., 1983*; *Li et al., 1996*; *Li et al., 1997*; *Travers and Norgren, 1983*), thereby, supporting our observation that the presynaptic inputs from MRN and dMRf to the hypoglossal and trigeminal motor nuclei observed in neonatal mice are lost in adult mice. In the case of vibrissa premotor tracing, no studies showed premotor labeling in the rostral brain areas such as EA and ZI. Future studies combining anterograde and retrograde tracers and causal manipulations will answer the function of these presynaptic inputs for whisking.

Ultimately, how orofacial premotor circuits coordinate different actions to produce synergistic orofacial behaviors will be of great interest. Such a fine circuit dissection will be enabled by future studies with multicolor RV tracing using additional sets of recombinase and receptor-virus envelope, such as AAV2retro-FlpO and TVB-EnvB (*Matsuyama et al., 2015*), which allow simultaneous tracing of premotor circuits from two different motoneuron groups involved in the same orofacial motor actions. We envision that transsynaptic premotor circuit tracing combined with functional characterization and activity manipulations will define the computational logic of orofacial motor control.

## Materials and methods

### Animals
All animal experiments were conducted according to protocols approved by the Duke University Institutional Animal Care and Use Committee.

Male and female C57B/L6 and Gt(Rosa)26Sor$^{tm14(CAG-tdTomato)Hze/J}$ (Ai14) (JAX # 007914) mice were obtained from the Jackson Laboratory (Bar Harbor, ME, USA) and used for virus tracing experiments.

### Viruses
AAV2retro-CAG-Cre ($2 \times 10^{12}$ gc/ml, Boston Children's Hospital Viral Core) (*Tervo et al., 2016*). AAV2/8-CAG-Flex-TVA-mCherry ($1.7 \times 10^{13}$ vg/ml, #48332, Addgene, Cambridge, MA, USA) (*Miyamichi et al., 2013*). AAV2/8-CAG-Flex-oG ($1 \times 10^{13}$ vg/ml, #74292, Addgene) (*Kim et al., 2016*). EnvA(M21)-RV-ΔG-GFP (also called CANE-ΔG-RV, $5 \times 10^8$ ifu/ml) (*Sakurai et al., 2016*; *Wickersham et al., 2007*). Retrograde lentivirus (*Kato et al., 2011*) carrying CreN and CreC (RG-LV-CreN, RG-LV-CreC, $1 \times 10^8$ ifu/ml) (*Stanek et al., 2016*; *Wang et al., 2012*).

### Monosynaptic transsynaptic rabies virus tracing
The tracing was performed in three steps.

#### (1) Peripheral tissue injection
To label a specific group of orofacial motor neurons, AAV2-retro-CAG-Cre (1000 nl) was injected into either the mystacial pad, genioglossus, or masseter muscles at postnatal day 17 using a volumetric injection system (based on a single-axis oil hydraulic micromanipulator MO-10, Narishige International USA, Inc, East Meadow, NY, USA) (*Petreanu et al., 2009*) equipped with a pulled and beveled glass pipette (Drummond, 5-000-2005). Before injection, mice were anesthetized by a cocktail of ketamine and xylazine (100 mg/kg and 10 mg/kg, i.p.). For the mystacial pad, the virus was injected subcutaneously into the areas around C2 and B2 vibrissae (500 nl each). For the genioglossus, the virus was injected directly into the muscle after exposing it by ventral neck dissection. Briefly, the genioglossus muscle was exposed by making a small incision in the mylohyoid muscle after the anterior digastric muscle was split open in the midline. For the masseter, the virus was injected into the area between the buccal and marginal nerves after making a small incision on a skin.

#### (2) Helper virus injection
For specific infection and glycoprotein complementation of pseudotyped RV-ΔG, helper viruses (120 nl, 1:1 mixture of AAV2/8-CAG-Flex-TVA-mCherry and AAV2/8-CAG-Flex-oG) were stereotaxically injected into the lateral part of the facial motor, hypoglossus, or trigeminal motor nuclei using a stereotaxic instrument (model 963, David Kopf Instruments, Tujunga, CA, USA) 3 weeks or longer

after the peripheral tissue injection. The viruses were injected at the rate of 30 nl/min with the injection system described above. The stereotaxic coordinates used were for the lateral part of the FN: 5.8 mm posterior, 1.38 mm lateral to the Bregma, and 5.2 mm below the brain surface; for the hypoglossus nucleus: 5.8 mm posterior, 0.05 mm lateral to the Bregma, and 5.15 mm below the brain surface with an anteroposterior 20° angle from vertical; and for the trigeminal motor nucleus: 4.1 mm posterior, 1.27 mm lateral to the Bregma, and 4.6 mm below the brain surface with an anteroposterior 20° angle from vertical. Before suturing the skin, the craniotomy was filled with kwik-sil (World Precision Instruments, Inc, Sarasota, FL, USA) and covered with cyanoacrylate glue (Super Glue, Loctite, Westlake, OH, USA).

### (3) Pseudotyped RV injection
Two weeks after the helper virus injection, EnvA(M21)-RV-ΔG-GFP (250 nl) was stereotaxically injected into the lateral part of the facial motor, hypoglossus, or trigeminal motor nuclei as described above.

## Retrograde split-Cre tracing
To label premotor neurons innervating multiple distinct motor nuclei, retrograde lentivirus carrying CreC or CreN (RG-LV-CreN and RG-LV-CreC) was stereotaxically injected separately into target motor nuclei of Cre-dependent tdTomato reporter mice. Specifically, for $VII_{middle}$-XII premotor neurons, RG-LV-CreN (750 nl) and RG-LV-CreC (500 nl) were injected into $VII_{middle}$ (5.8 mm posterior, 1.3 mm lateral to the Bregma, and 5.2 mm below the brain surface) and the hypoglossus nucleus, respectively. For bilateral $VII_{middle}$ premotor neurons, RG-LV-CreN (750 nl) and RG-LV-CreC (500 nl) were injected into left and right $VII_{middle}$, respectively.

## Histology
Five days after the pseudotyped RV injection, the animals were deeply anesthetized with isoflurane and transcardially perfused with 10% sucrose in Milli-Q water, followed by ice-cold 4% paraformaldehyde in 0.1 M phosphate buffer, pH 7.4. After dissection, the brains were post-fixed in the same fixative for overnight at 4°C and freeze-protected in 30% sucrose in phosphate buffer saline (PBS) at 4°C until they sank. The brains were embedded in OCT compound (Sakura Finetek USA, Inc, Torrance, CA, USA) and frozen in dry-ice-cooled ethanol. 80 µm free-floating coronal sections were made using a cryostat (Leica Biosystems Inc, Buffalo Grove, IL, USA). The sections were briefly washed in PBS and stained with Neurotrace blue fluorescent Nissl stain (1:500, Thermo Fisher Scientific, Waltham, MA, USA) in 0.3% Triton-X100/PBS for overnight at 4°C. The sections were briefly washed and mounted on slide glasses with Mowiol.

For retrograde split-Cre tracing experiments, some sections were stained with rabbit anti-RFP (1:500, #600-401-379, Rockland Immunochemicals, Inc, Limerick, PA, USA) and goat anti-choline acetyltransferase (1:500, #AB144P, MilliporeSigma, Burlington, MA, USA) antibodies. Primary antibodies were visualized using donkey anti-rabbit antibody conjugated with Alexa Fluor Plus 555 (1:1000, #A32794, Thermo Fisher Scientific) and donkey anti-goat antibody conjugated with Alexa Fluor Plus 488 (1:1000, #A32814, Thermo Fisher Scientific).

## Imaging
Fluorescent images for atlas registration were taken with a Zeiss 700 laser scanning confocal microscope (Carl Zeiss Inc, Thornwood, NY, USA) using a ×10 objective (pixel size, 1.042 × 1.042 µm).

## Mapping of labeled neurons in Allen CCF
A previously published method SHARP-track (*Shamash et al., 2018*) was modified to improve registration of the brainstem sections to Allen CCF. Briefly, three steps for the registration were either introduced or improved, including (i) user-assist nonrigid deformation registration, (ii) correct conversion of coordinates (ML and DV to Bregma) after diffeomorphic registering of a section to the Allen CCF along the AP axis, and (iii) an option for automatic cell identification. First, the affine transform used by SHARP-track for brain section to reference registration is upgraded to LogDemons methods (*Lu et al., 2018*), a fast diffeomorphic registration method that can handle a more diverse scenario of section distortion. Second, the procedures to determine the coordinates of the brain sections after nonrigid transformation and registration were corrected, and this is also critical for three-dimensional

reconstruction and visualizations of the results from serial two-dimensional sections. Third, in addition to the manual cell identification by user-generated click, an optional automatic cell identification function was developed to recover most identifiable cells, and subsequently users can manually correct mistakes. The automatic cell identification method contains a series of simple filters that balanced the speed and the precision of the approximation. Detailed implementation can be found in the GitHub repository (https://github.com/wanglab-neuro/Allen_CCF_reg, *Takatoh, 2021*). This site will be freely available upon publication. Coordinates of Bregma in Allen CCF were set at AP, 5400; ML, 5700; DV 0 (*Shamash et al., 2018*).

## Spatial correlation analysis

The cells detected in each mouse were first registered into the standard three-dimensional brain model. The (x, y, z) coordinates of each cell were then extracted, and the multivariate kernel smoothing density estimation was applied (bandwidth = 1). The resulting kernel density estimation was then vectorized, and the cosine similarity between any two of the mice was calculated to form the correlogram. The result was shaped to a square matrix, which is shown in the figure. The code for this analysis can be requested from the corresponding authors.

## Visualization of labeled neurons on Allen CCF

Premotor neurons registered in Allen CCF were visualized using Brainrender (*Claudi et al., 2021*) or a custom written code (for density plots). Briefly, the coordinates are converted into Allen CCF coordinates by multiplying 1000 and adding 5400 (for AP) and 5700 (for ML). Converted cell coordinates were plotted using Points function in Brainrender v2.0.0.0 by following the instruction (https://github.com/brainglobe/brainrender). Density plots are generated by using kdeplot function in seaborn (https://seaborn.pydata.org).

## Acknowledgements

We thank Lauren E McElvain and Harvey J Karten for discussions on anatomical annotation and Martin Deschênes for discussions on the viral labeling strategy. We thank members of the Wang laboratory for helpful discussions and suggestions over the course of this work. This work was supported by NIH grants U19 NS107466 and R01 NS077986.

---

## Additional information

### Funding

| Funder | Grant reference number | Author |
| --- | --- | --- |
| National Institute of Neurological Disorders and Stroke | NS107466 | David Kleinfeld Fan Wang |
| National Institute of Neurological Disorders and Stroke | NS077986 | Fan Wang |

The funders had no role in study design, data collection and interpretation, or the decision to submit the work for publication.

### Author contributions

Jun Takatoh, Conceptualization, Data curation, Formal analysis, Investigation, Methodology, Software, Validation, Visualization, Writing – original draft, Writing – review and editing; Jae Hong Park, Data curation, Investigation, Methodology, Validation, Writing – review and editing; Jinghao Lu, Formal analysis, Methodology, Software, Validation, Visualization, Writing – review and editing; Shun Li, Investigation, Validation; PM Thompson, Data curation, Software, Validation; Bao-Xia Han, Shengli Zhao, Resources; David Kleinfeld, Conceptualization, Funding acquisition, Supervision; Beth Friedman, Methodology, Validation, Writing – review and editing; Fan Wang, Conceptualization, Funding acquisition, Methodology, Supervision, Writing – original draft, Writing – review and editing

## Author ORCIDs

Jun Takatoh [iD] http://orcid.org/0000-0002-0976-7684
Shun Li [iD] http://orcid.org/0000-0002-8580-3843
PM Thompson [iD] http://orcid.org/0000-0001-6083-8831
David Kleinfeld [iD] http://orcid.org/0000-0001-9797-4722
Fan Wang [iD] http://orcid.org/0000-0003-2988-0614

## Ethics

Animal experimentation: All experiments were conducted according to protocols approved by the Duke University Institutional Animal Care and Use Committee protocol (# A143-18-06).

## Decision letter and Author response

Decision letter https://doi.org/10.7554/eLife.67291.sa1
Author response https://doi.org/10.7554/eLife.67291.sa2

---

# Additional files

## Supplementary files

• Transparent reporting form

## Data availability

All data generated or analysed during this study are included in the manuscript and supporting files.

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
