## [Decision Letter]

**Acceptance summary:**

Facial muscles control the execution of essential tasks like eating, drinking, breathing and (in most mammals) tactile exploration. The activity of motor neurons targeting different muscles are coordinated by premotor regions distributed throughout brainstem. The precise identity of these cells and regions in adults is presently unclear, largely due to technical challenges. Here, the authors make use of a 3-way viral strategy to visualize of premotor neurons in the adult and align the anatomical data to a common reference brain. The work offers insight into the dynamics of premotor circuit distribution between development and adult.

**Decision letter after peer review:**

Thank you for submitting your article "Constructing An Adult Orofacial Premotor Atlas In Allen Mouse CCF" for consideration by *eLife*. Your article has been reviewed by 3 peer reviewers, including Alexander Theodore Chesler as the Reviewing Editor and Reviewer #1, and the evaluation has been overseen by Catherine Dulac as the Senior Editor.

Essential revisions:

This work represents a significant advance over previous findings in that the new method developed is applied to adult mice. The authors find premotor circuits in adult mice that closely match those in neonatal mice, with important exceptions. This work advances our knowledge by revealing orofacial premotor circuits in adults and the reviewers feel it is suitable to publication in *eLife* with a few additional pieces of experimental data and additions/revisions to the text.

*Reviewer #1 (Recommendations for the authors):*

Technically the study is well founded but there are some issues:

1. The innovation revolves around the new method for retrograde labeling, but the authors rather arbitrarily chose P17 injection of muscles, a time when at least some refinement is incomplete (sup 1). Do the data reflect adult patterns or an intermediate state for all or some of the muscle groups?

2. 3 injections are required raising questions of efficiency. Unfortunately, because founder cells for rabies tracing were not observed, there is no control for this at any level.

3. The split Cre is a nice extension to Figure 7 but does not make use of muscle tracing but rather the generalized literature assignment of regions of the different motor nuclei for starter targeting. If this is valid, why is the complex 3 virus approach needed?

*Reviewer #2 (Recommendations for the authors):*

The paper is suitable for publication in eLife with a few revisions.

1. The authors should specify both volumes and titers of viruses used in this study.

2. Data for specificity of infection by retrograde injection should be provided for all three motoneuron pools. It might be that that time lines are distinct for the three different pools and hence this should be analysed and shown to make sure the timing of injection is fine with respect to specificity for all three datasets. Related, the time point P17, which is the one the authors use is not shown in the Suppl Figure.

3. For masseter injections, there appears quite a bit of variability for different animals in terms of premotor distribution (Figure Supplement 5B). Is there any way to check injection specificity (see also point 2)?

4. Dual projecting neurons are likely a minor population since the peak of the cluster of premotor neurons for the different starter motoneuron pools is described to be distinct. The split-Cre approach does not allow to make the claim about premotor neurons. It has the same caveat as the authors mention in the introduction for conventional tracers. The authors should make these points clear in the discussion, although quantification is not possible with the available datasets.

5. It is difficult to understand the correlation plot. The authors should provide details and the script.

*Reviewer #3 (Recommendations for the authors):*

– Lines 232-234: the reference to the MeV figure panel should be Figure 4F, not Figure 4E. The reference to the DCN panel should be Figure 4G.

– Figure supplement 8, panel B: the two circles representing injection sites are labeled "VII" and "XII", but both should be "VII".

– Line 124: my understanding of "systemic" means via the bloodstream. Is this what's meant?

– Line 469: The Stanek reference is missing a year.

---

## [Author Response]

Reviewer #1 (Recommendations for the authors):

Technically the study is well founded but there are some issues:

1. The innovation revolves around the new method for retrograde labeling, but the authors rather arbitrarily chose P17 injection of muscles, a time when at least some refinement is incomplete (sup 1). Do the data reflect adult patterns or an intermediate state for all or some of the muscle groups?

The reviewer is right that we do not know whether by P17, neuromuscular innervation has been fully refined from polyneuronal innervation to single motor-axon-muscle fiber innervation for these muscles.

However, motoneuron specification and pathfinding to muscle targets (i.e., the specific muscle group) happen during later embryonic development (Ashwell and Watson, 1983; Chen et al., 2016; Tenney et al., 2019). Thus, the retrogradely labeled motoneurons are specific for the target muscle group, even if some of them might have residual innervation of more than one muscle fibers of the target muscle. Since we do not intend to trace premotor neurons for a single motoneuron/single muscle fiber, we do not think the neuromuscular junction refinement would be a concern here.

Additionally, we inject helper AAV later than P27, so the premotor circuits we traced are mature circuits.

2. 3 injections are required raising questions of efficiency. Unfortunately, because founder cells for rabies tracing were not observed, there is no control for this at any level.

We have now provided a supplemental figure showing the “starter cells” including “dying starter cells”, in the respective motor nuclei (Figure supplement 3A, 3B (facial motor nucleus); 4A, 4B (hypoglossal nucleus); 5A, 5B (trigeminal motor nucleus)). We have mapped these motoneurons also onto Allen CCF.

3. The split Cre is a nice extension to Figure 7 but does not make use of muscle tracing but rather the generalized literature assignment of regions of the different motor nuclei for starter targeting. If this is valid, why is the complex 3 virus approach needed?

We should have explained more.

So, prior to the availability of AAVretro, we have tried to use retrograde lentivirus (RG-LV) or muscle injection to retrogradely label specific motoneurons. However, RG-LV turned out to be inefficient at infecting motoneurons from muscles, hence the 3-step strategy depends on the efficiency of AAVretro to infect specific motoneurons innervating a specific muscle group.

Interestingly, we tried AAVretro injection into various motor nuclei, and was surprised to see that AAVretro has tropism issues of infecting brainstem premotor neurons (Data not shown). Thus, even though the RG-LV is not as efficient as AAVretro at infecting neurons, it at least can generally infect brainstem neurons as we have shown before (Bellavance et al., 2017; Stanek et al., 2016).

The goal of the RG-LV-split-Cre experiment is then to find possible sources of common premotor neurons innervating two (or more) spatially separate motor nuclei, and in combination with our monosynaptic tracing, we could then know which premotor regions contain neurons innervating multiple groups of motoneurons. We should point out that the RG-LV-split-Cre strategy by itself is not specific. Thus, only in reference to the existing premotor neuron maps, it can be used to tell us where the “common” premotor neurons are. We have now added the caveats of the split-Cre strategy to Discussion: Line 482-488.

Reviewer #2 (Recommendations for the authors):

The paper is suitable for publication in eLife with a few revisions.

1. The authors should specify both volumes and titers of viruses used in this study.

We have now provided volumes and titers of viruses in Materials and Methods.

2. Data for specificity of infection by retrograde injection should be provided for all three motoneuron pools. It might be that that time lines are distinct for the three different pools and hence this should be analysed and shown to make sure the timing of injection is fine with respect to specificity for all three datasets. Related, the time point P17, which is the one the authors use is not shown in the Suppl Figure.

We have now provided a supplemental figure showing the “starter cells” in the respective motor nuclei (Figure supplement 3A, 3B (facial motor nucleus); 4A, 4B (hypoglossal nucleus); 5A, 5B (trigeminal motor nucleus)), and have mapped them to Allen CCF as well.

3. For masseter injections, there appears quite a bit of variability for different animals in terms of premotor distribution (Figure Supplement 5B). Is there any way to check injection specificity (see also point 2)?

Yes, the source cells are now shown and mapped.

The masseter muscle is the largest craniofacial muscle with a long-stretching band of neuromuscular junctions, and most of the trigeminal (V) motor nucleus are composed of masseter motoneurons. We speculate that the masseter premotor neurons are also quite extensive and spread out in brainstem, and thus depending the initial source infected masseter motoneurons, there is likely more variation in labeled premotor neurons than that seen for whisker and genioglossus motoneurons.

4. Dual projecting neurons are likely a minor population since the peak of the cluster of premotor neurons for the different starter motoneuron pools is described to be distinct. The split-Cre approach does not allow to make the claim about premotor neurons. It has the same caveat as the authors mention in the introduction for conventional tracers. The authors should make these points clear in the discussion, although quantification is not possible with the available datasets.

Agreed. Reviewer 1 also has comments on the split-Cre strategy. We have now added the caveats in Discussion, Line 482-488

5. It is difficult to understand the correlation plot. The authors should provide details and the script.

The cells detected in each mouse were first registered into the standard three-dimensional brain model. The (x, y, z) coordinates of each cell were then extracted, and the multivariate kernel smoothing density estimation was applied (bandwidth = 1). The resulting kernel density estimation was then vectorized, and the cosine similarity between any two of the mice were calculated to form the correlogram.

We have now added these descriptions to the Methods. The codes can be requested from corresponding authors.

Reviewer #3 (Recommendations for the authors):

– Lines 232-234: the reference to the MeV figure panel should be Figure 4F, not Figure 4E. The reference to the DCN panel should be Figure 4G.

It is now corrected to Figure 4F (MeV) and Figure 4G (DCN)

– Figure supplement 8, panel B: the two circles representing injection sites are labeled "VII" and "XII", but both should be "VII".

It is now corrected. Also, Figure supplement 8b has been reorganized to make it easier to understand.

– Line 124: my understanding of "systemic" means via the bloodstream. Is this what's meant?

Yes. It is now rewritten as “likely through entering the fluid of the peri-nerve space after injection” (Line 132-133). We think that because myelination and other connective tissues surrounding the nerve are immature in neonatal animals, and different branches of the facial nerve come together to the main nerve, the virus could infect other facial motoneurons when they travel up to the nerve. But as the animal mature, maturation of connective tissues and myelination will prevent such spill over spreading.

– Line 469: The Stanek reference is missing a year.

The reference is inserted.

References:

Ashwell, K.W., and Watson, C.R. (1983). The development of facial motoneurones in the mouse – neuronal death and the innervation of the facial muscles. J Embryol Exp Morphol *77*, 117-141.

Bellavance, M.A., Takatoh, J., Lu, J., Demers, M., Kleinfeld, D., Wang, F., and Deschenes, M.

(2017). Parallel Inhibitory and Excitatory Trigemino-Facial Feedback Circuitry for Reflexive Vibrissa Movement. Neuron *95*, 673-682 e674, doi: 10.1016/j.neuron.2017.06.045.

Chen, X., Wang, J.W., Salin-Cantegrel, A., Dali, R., and Stifani, S. (2016). Transcriptional regulation of mouse hypoglossal motor neuron somatotopic map formation. Brain Struct Funct

*221*, 4187-4202, doi: 10.1007/s00429-015-1160-2.

Stanek, E., 4th, Rodriguez, E., Zhao, S., Han, B.X., and Wang, F. (2016). Supratrigeminal

Bilaterally Projecting Neurons Maintain Basal Tone and Enable Bilateral Phasic Activation of Jaw-Closing Muscles. J Neurosci *36*, 7663-7675, doi: 10.1523/JNEUROSCI.0839-16.2016.

Tenney, A.P., Livet, J., Belton, T., Prochazkova, M., Pearson, E.M., Whitman, M.C., Kulkarni, A.B., Engle, E.C., and Henderson, C.E. (2019). Etv1 Controls the Establishment of Nonoverlapping Motor Innervation of Neighboring Facial Muscles during Development. Cell Rep*29*, 437-452 e434, doi: 10.1016/j.celrep.2019.08.078.